# Impact of Quercetin against *Salmonella* Typhimurium Biofilm Formation on Food–Contact Surfaces and Molecular Mechanism Pattern

**DOI:** 10.3390/foods11070977

**Published:** 2022-03-28

**Authors:** Pantu Kumar Roy, Min Gyu Song, Shin Young Park

**Affiliations:** Department of Seafood Science and Technology, Institute of Marine Industry, Gyeongsang National University, Tongyeong 53064, Korea; vetpantu88@gmail.com (P.K.R.); mg8117@naver.com (M.G.S.)

**Keywords:** *Salmonella* Typhimurium, antioxidant, quercetin, biofilm, rubber, hand gloves, gene expression

## Abstract

Quercetin is an active nutraceutical element that is found in a variety of foods, vegetables, fruits, and other products. Due to its antioxidant properties, quercetin is a flexible functional food that has broad protective effects against a wide range of infectious and degenerative disorders. As a result, research is required on food-contact surfaces (rubber (R) and hand gloves (HG)) that can lead to cross-contamination. In this investigation, the inhibitory effects of quercetin, an antioxidant and antibacterial molecule, were investigated at sub-MIC (125; 1/2, 62.5; 1/4, and 31.25; 1/8 MIC, μg/mL) against *Salmonella* Typhimurium on surfaces. When quercetin (0–125 μg/mL) was observed on R and HG surfaces, the inhibitory effects were 0.09–2.49 and 0.20–2.43 log CFU/cm^2^, respectively (*p* < 0.05). The results were confirmed by field emission scanning electron microscopy (FE-SEM), because quercetin inhibited the biofilms by disturbing cell-to-cell connections and inducing cell lysis, resulting in the loss of normal cell morphology, and the motility (swimming and swarming) was significantly different at 1/4 and 1/2 MIC compared to the control. Quercetin significantly (*p* < 0.05) suppressed the expression levels of virulence and stress response (*rpoS*, *avrA*, and *hilA*) and quorum-sensing (*luxS*) genes. Our findings imply that plant-derived quercetin could be used as an antibiofilm agent in the food industry to prevent *S.* Typhimurium biofilm formation.

## 1. Introduction

Foodborne diseases are considered one of the major public health problems in the world today, producing a significant rate of morbidity and mortality [1,2]. *Salmonella* spp. is a type of bacteria that can cause food poisoning. According to the Centers for Disease Control and Prevention (CDC), Salmonellosis causes approximately 1.35 million infections, 26,500 hospitalizations, and 420 deaths in the USA annually [3]. The incidence of salmonellosis has grown because of the presence of salmonella spp. biofilm on numerous food-contact surfaces in poultry and chicken processing plants [4].

Biofilm is a bacterial structure population that is permanently adherent to biotic and abiotic surfaces and incorporated in a self-produced extracellular polymeric matrix [5,6]. Biofilm production on foods and food-contact surfaces results in contamination, post-process contamination, and cross-contamination of the final product, resulting in food spoiling, product rejection, economic losses, and foodborne diseases [4,7,8]. Bacteria in biofilms are more adaptable to varied environmental conditions than planktonic cells. As a result, biofilm, which is difficult to remove, is a significant hygiene issue in the food industry [9]. Food spoilage and pathogenic bacteria have ability to adhere to food-processing surfaces, such as stainless steel (SS), silicon rubber (SR), plastic (PLA), rubber gloves (RG), and food surfaces, and biofilm formation is a serious public health concern since resistant biofilms can be a persistent source of contamination [10,11].

Quorum-sensing (QS) means that bacteria monitor their population density and release signaling molecules called autoinducers to communicate each other [8]. QS controls virulence factors, extracellular enzymes (e.g., proteases, pectinases, and lipases), biofilm formation, secondary metabolites, and motility, among other physiological activities in bacteria [5]. *RpoS* is a stress response protein that regulates the expression of genes linked to resistance to various environmental challenges (such as hunger, oxidative damage, and low pH), and environmental stress is linked to biofilm formation [12]. *AvrA* and *hilA* are virulence genes associated with invasiveness in Salmonella pathogenicity islands-1 (SPI-1) [4]. *LuxS* is associated with the quorum-sensing through the synthesis of AI-2 (autoinducer-2) molecules [8,13]. Traditional biofilm treatment is becoming increasingly ineffective due to bacterial resistance, which is a serious issue contributing to food degradation and rejection due to the danger of unfit human ingestion [14]. As a result, great effort has been expended recently to develop preventive measures that target the initial adhesion phase or impede intercellular communication while posing no hazard to the presence of bacteria [15]. QS modulates various phenotypes in bacteria, coordinating a group behavior that governs virulence factors, extracellular enzymes, biofilm formation, secondary metabolites, and motility, among other things [16,17]. Many of these traits can cause food to deteriorate, making the product unfit for human consumption [18]. As a result, researchers have been working on ways to use inhibitory chemicals to disrupt this communication and improve food quality and safety [16,17]. Many studies have revealed that plant organic extracts high in phenolic compounds can interfere with QS in various bacteria. These compounds are a varied set of chemical molecules, with various chemical actions vital for plant reproduction, development, and pathogen defense [17,19]. They are divided into phenolic acids, stilbenes, lignans, and flavonoids, based on the number of rings and the type of component that bind those [20]. Flavonoids are a significant category of natural products with a polyphenol structure that can be found in several fruits and vegetables [21,22,23]. Flavonoids are popular due to their anti-inflammatory, antibacterial, antioxidant, and anticancer qualities, among other things [24,25]. Furthermore, flavonoids have gained prominence as possible inhibitors of the QS system. Taxifolin, kaempferol, naringenin, apigenin, baicalein, and other flavonoids interfere with the QS system of bacteria like *Pseudomonas aeruginosa* PAO1 and *Chromobacterium violaceum* CV026 [17,26], affecting the transcription of QS-controlled target promoters and limiting the synthesis of virulence factors [27].

Quercetin is a flavonoid-based chemical found in apples, tea, onions, wine, red grapes, berries, and tomatoes, among other plants and fruits [28]. It has many uses, including antioxidant, free radical scavenger, anticancer, and neuroprotective properties [29,30]. Due to its three-ring structure with 5 hydroxyl groups, it possesses exceptionally powerful antioxidant capabilities [30]. Antioxidants reduce oxidative stress and limit biofilm development by eliminating reactive oxygen species (ROS) accumulated in bacterium cells, making them efficient antibiofilm agents [30,31]. Additionally, the antibacterial effect of quercetin against Gram-positive [30,32] (e.g.: *Staphylococcus aureus*) and Gram-negative bacteria [33,34] (e.g.: *Escherichia coli* and *P. aeruginosa*) had already been studied. However, there has been insufficient research on quercetin antibiofilm impact against *Salmonella* Typhimurium. Therefore, the efficiency of quercetin at sub-minimum inhibitory concentration (sub-MIC) against QS-regulated behavior of *S.* Typhimurium, including biofilm formation and flagellar-mediated motility, as well as its impact on the virulence, stress response, and QS gene expression were investigated in the present study.

## 2. Materials and Methods

### 2.1. Bacterial Strains and Culture Conditions

*Salmonella* Typhimurium (ATCC 14028) was used in this study. The bacterial strains (cell density: 10^8^–10^9^ CFU/mL) were stored in a deep freezer at −80 °C with phosphate buffered saline (PBS; Oxoid, Basigstoke, England) containing 30% glycerol in stock vial. First, 100 μL of bacteria was inoculated into the 10 mL tryptic soy broth (TSB; BD Dicfo, Franklin Lakes, NJ, USA) and stored at 37 °C and 200 rpm shaking incubator (Vision Scientific, VS–8480, Gyeongsan, South Korea). After 24 h, 100 μL was taken from the culture medium and inoculated in 10 mL fresh TSB. Then, it was stored in a shaking incubator under the same conditions as the previous day. After 18 h, the culture medium was centrifuged for 10 min at 10,000 rpm and 4 °C, and washed with PBS at least two times. Then, the final bacterial solution was diluted with peptone water (PW; Oxoid, Basigstoke, England) until the number of bacteria in the solution was 10^5^ log CFU/mL.

### 2.2. Preparation of Quercetin

Quercetin (Q–4951) was purchased from Sigma-Aldrich (St. Louis, MO, USA). The product was used after dissolving in dimethyl sulfoxide (DMSO, Sigma-Aldrich, St. Louis, MO, USA) and made stock solution concentration 1 mg/mL.

### 2.3. Preparation of Samples (Food-Contact Surfaces)

For food contact surfaces, hand gloves latex (HG; Komax industrial Co., Ltd., Seoul, Korea) and rubber coupons (2 × 2 × 0.5 cm) (R; Ultra High Modular Weight Polyethylene; JINIL Tec-PLA Co., Seoul, Korea) were used. Hand gloves were cut into 2 × 2 cm using sterilized scissors. All coupons (R and HG) were dipped into 70% ethanol overnight and dried in UV light (Sankyo UV Co., Ltd., Seoul, Korea) until the ethanol was completely dried. Lastly, all samples were dried in 1000 μW/cm^2^ UV light for 15 min [10,35,36].

### 2.4. Determination of Minimum Inhibitory Concentration (MIC)

The MIC was confirmed, as previously described, with slight modifications [37]. To determine the minimum inhibitory concentration (MIC) of quercetin against *S.* Typhimurium, a two-fold serial dilution method using TSB was adopted. A 100 μL quercetin serially diluted with TSB and 100 μL bacterial suspension (10^5^ log CFU/mL) were mixed in 96-well plates (Corning Incorporated, Corning, Inc., Corning, NY, USA). Total volume was 200 μL in each well. The plates were stored in 37 °C incubator for 24 h, and absorbance (600 nm) was monitored with a microplate reader (Spectra Max 190, Sunnyvale, CA, USA).

### 2.5. Motility Assays

Motility assays in this study were performed as previously described, with slight modifications [4,38]. This experiment was performed to confirm the effect of quercetin on the two types of motility (swimming and swarming) of *S*. Typhimurium. Each medium used in the swimming and swarming experiments were made by adding 0.3% and 0.5% Bacto agar (BD Dicfo, Franklin Lakes, NJ, USA) to TSB, respectively. The medium was autoclaved and poured onto each plate. Before it hardened, quercetin was added, and mixed well carefully. In the swimming experiment, 4 μL of diluted bacterial suspension (10^5^ log CFU/mL) was inoculated by passing through a semi-solid medium. In the swarming experiment, 4 μL bacterial suspension was placed on the middle of the medium. All plates were stored in a 37 °C incubator, and the swimming plates were observed after 10 h and the swarming plates were observed after 33 h to measure the horizontal and vertical diameters.

### 2.6. Biofilm Formation and Detachment

This method was performed as previously described, with some modifications [4,8,10]. In this study, the MIC was 250 μg/mL, and the inhibitory effect of biofilm was observed at sub-MIC that may not kill the bacteria, but affect the virulence factor. The concentrations used in this study were control, 1/8, 1/4, and 1/2 MIC. The prepared samples were put into 50 mL conical tube containing 10 mL TSB, quercetin, and 100 μL of bacterial suspension (10^5^ log CFU/mL). Finally, they were mixed well using a vortex mixer (Scientific Industries, SI–0256, Bohemia, NY, USA) and incubated for 24 h at 37 °C. After the biofilm formation, the coupons were washed twice with distilled water (DW) to remove bacteria which slightly adhered to the surfaces. The washed coupons were immersed in 10 mL peptone water (PW) consisting of 10 sterilized glass beads, and then vortexed for 2 min. After serial dilution of this bacterial suspension, it was inoculated and spread on Xylose lysine deoxycholate (XLD) plates. After storing them in a 37 °C incubator for 24 h, the number of colonies on the plates were counted. Finally, we obtained the inhibition values by subtracting the population of each concentration (1/8, 1/4, and 1/2 MIC) from the population of each control group.

### 2.7. Field Emission Scanning Electron Microscopy (FE-SEM)

This experiment was performed as previously described, with some modifications [10,39]. FE-SEM was performed to visually confirm the effect of quercetin on the biofilm formation of *S.* Typhimurium on the surfaces of HG. Biofilms were formed on each coupon using the method already described in this paper. The samples were washed twice with PBS and fixed with 2.5% glutaraldehyde (Sigma-Aldrich, St. Louis, MO, USA) for 4 h. In the case of HG, the fixed samples were washed three times with PBS for 10 min, and each sample was serially treated in the order of 50, 60, 70, 80, 90, and 100% ethanol (100% ethanol was treated three times.). After the first dehydration, the second dehydration was performed using hexamethyldisilazane (HMDS; Sigma–Aldrich, St. Louis, MO, USA) at concentration of 25, 50, 75, and 100% diluted in ethanol (100% HMDS was treated three times). Then, each sample was stored in a desiccator for about 1 day. Finally, all the samples were coated with platinum (Pt) and observed with FE-SEM (Hitachi/Baltec, Hitachinaka, Japan). FE-SEM was taken with an acceleration voltage of 5 kV at working distances ranging from 7 to 10 mm [40].

### 2.8. RNA Extraction, cDNA Synthesis, and Real-Time PCR (RT–PCR) Analysis

This experiment was performed as previously described, with slight modifications [4]). It was performed to verify the effect of quercetin on the expression of virulence and quorum-sensing genes in *S*. Typhimurium. The bacteria (10^5^ log CFU/mL) were inoculated into each Falcon^®^ tube containing 10 mL TSB with quercetin. They were incubated for 24 h in a 37 °C incubator. After biofilm formation, total RNA was extracted from the pellet obtained by centrifugation, using the RNeasy Mini kit (Qiagen, Hilden, German). The RNA yield and purity were determined by a spectrophotometer at 260/280 nm and 260/230 nm (NanoDrop, Bio-Tek Instruments, Chicago, IL, USA), and then cDNA was synthesized using a Maxime RT PreMix (Random Primer) kit (iNtRON Biotechnology Co., Ltd., Seoul, Gyeonggi-do, Korea). The primers are shown in Table 1. The 16S rRNA was used as the housekeeping gene. Briefly, the complementary DNA sample was mixed with respective primers and Power SYBR Green PCR Master Mix (Applied Biosystems, Thermo Fisher Scientific, Warrington, UK) in a total volume of 20 µL. RT–PCR analysis was achieved using a CFX Real–Time PCR System (Bio–Rad, Hercules, CA, USA). RT–qPCR was performed using 1 μL of cDNA as a template and 2X Real–Time PCR Master Mix. Real-time PCR was performed using a CFX Real-Time PCR System (Bio–Rad, Hercules, CA, USA). The PCR reaction protocol started with initial denaturation at 95 °C for 20 s, 50 °C for 20 s, and 72 °C for 20 s, respectively [41,42,43]. After the completion of PCR cycling, we acquired melting curves to verify the specificity and analyzed by 2^−^^△△Ct^ method [44,45,46].

### 2.9. Statistical Analysis

All experiments were repeated at least three times. All data were expressed as mean ± standard error of mean (SEM). The significance was determined by Ducan’s multiple–range test and ANOVA using SAS software version 9.2 (SAS Institute Inc., Cary, NC, USA), and statistical significance was set at *p* < 0.05.

## 3. Results

### 3.1. Determination of Minimum Inhibitory Concentration (MIC) and Sub-MIC

The MIC of quercetin against *S*. Typhimurium was 250 μg/mL. In the present study, the inhibitory effect of sub-inhibitory concentrations of quercetin (125, 62.5, and 31.25 μg/mL) on *S*. Typhimurium motility, biofilm formation, virulence, stress response, and quorum-sensing gene expression were investigated.

### 3.2. Motility Assays (Swimming and Swarming)

The mobility of bacterial flagella is critical for biofilm formation. Swimming and swarming tests, in particular, can confirm the flagella mobility of *Salmonella* spp. Figure 1 and Figure 2 indicate the effect of quercetin on *S.* Typhimurium motility inhibition. The swimming experiment revealed that quercetin inhibited the motility of *S*. Typhimurium by 16% and 76% compared to control at 1/8 and 1/2 MIC, respectively. The inhibitory effect of quercetin on *S*. Typhimurium is shown in Figure 2. As a result, quercetin inhibited the motility of *S.* Typhimurium by 12% and 54.5% at 1/8 and 1/2 MIC, respectively. Therefore, in this experiment, swimming and swarming motility were more inhibited as the concentration of quercetin increased. Particularly, 1/2 MIC of quercetin showed a significant (*p* < 0.05) difference in motility compared to the control group.

### 3.3. Inhibitory Effect of Quercetin against S. Typhimurium Biofilm on Food-Contact Surfaces (R and HG)

Figure 3 shows quercetin inhibitory effect on *S.* Typhimurium biofilm on R coupons. Biofilm inhibitory effect increased with quercetin concentration. When quercetin concentrations were 1/8, 1/4, and 1/2 MIC, the inhibitory values of *S.* Typhimurium biofilm on R surface were 0.09, 0.87, and 2.49 log CFU/cm^2^, respectively. As a result, these values were significantly (*p* < 0.05) inhibited at 1/2 MIC, compared to the control and other MIC groups. Figure 4 shows quercetin inhibitory effects on *S.* Typhimurium biofilm on HG. When the concentrations of quercetin were 1/8, 1/4, and 1/2 MIC, the inhibitory values of *S*. Typhimurium biofilm were 0.20, 0.79, and 2.43 log CFU/cm^2^, respectively. Biofilm inhibition by 1/2 MIC was significantly different when compared to the control and other MIC groups (*p* < 0.05).

### 3.4. Field Emission Scanning Electron Microscopy (FE-SEM)

The inhibitory effect of quercetin on the *S*. Typhimurium biofilm on the HG surface was visually confirmed with FE-SEM, and the results are shown in Figure 5. In particular, the inhibition of biofilm was much greater when exposed to quercetin at its ½ MIC than 1/8 MIC (Figure 5). As a result, in this experiment, the inhibitory effect of quercetin on the *S*. Typhimurium biofilm was visually confirmed.

### 3.5. Virulence, Stress Response, and Quorum-Sensing Gene Expression

The expression of virulence and stress response factors (*rpoS*, *avrA*, and *hilA*) and quorum-sensing factor *(luxS*) of *S.* Typhimurium, measured using real-time PCR in the presence of sub-inhibitory concentrations of quercetin (from 0 to 125 µg/mL), are shown in Figure 6. Gene expression was significantly reduced at the different levels of sub-MIC of quercetin (*p* < 0.05).

## 4. Discussion

Salmonellosis is a common food poisoning disease caused by *Salmonella* spp. that affects millions of people worldwide, and it is a serious problem in poultry meat and processing industries. Foodborne bacteria persistence in food processing environments is the principal source of food contamination, resulting in substantial issues and significant financial losses for the food industry. Due to concerns regarding the innocuity of some synthetic food preservatives, natural substances are quickly replacing chemical-based sanitizers and disinfectants [11]. Quercetin is present in plant extracts that could be considered as food ingredients rather than food additives. Quercetin is a nonspecific protein kinase enzyme inhibitor. In 2010, the FDA acknowledged high-purity quercetin as GRAS for use as an ingredient in various specified food categories, at levels up to 500 milligrams per serving. The current study investigated whether sub-MIC concentrations of quercetin may be used to reduce *S*. Typhimurium biofilm formation. The MIC of quercetin for *S.* Typhimurium was 250 μg/mL. Other authors [45] already reported that sub-inhibitory concentrations of antimicrobial molecules might reduce pathogenicity of bacteria, but nothing on their growth.

The motility of bacteria in a water-soluble or low viscosity state is referred to as swimming [46]. On a semi-solid surface, swarming is the movement of a bunch of cells [46]. Swarming has been linked to the production of bacterial biofilms, which are major virulence factors [47]. In this study, swimming and swarming motility were significantly reduced (*p* < 0.05) when quercetin concentration was 1/2 MIC, compared to control and other groups of MIC (Figure 1 and Figure 2). Other authors [48] reported quercetin of ½ MIC reduced motility, compared to the control group, which is related with our present study.

Our study exposed antibiofilm effects of quercetin against *S.* Typhimurium on different surfaces. The higher the concentration, the greater the inhibitory impact of quercetin. Other authors [49] reported a concentration of 0.2 mM was chosen to investigate the mode of action of quercetin against *Listeria monocytogenes* biofilm formation, because it allowed for generation of biofilm, which was required for the observation of modifications caused by quercetin at early and late stages of growth. Furthermore, the effect of quercetin (0.2 mM) on *L. monocytogenes* planktonic growth kinetics were evaluated to rule out any effect of this drug on planktonic populations during the experiment. Planktonic cells in the bulk medium contribute to increased biofilm thickness during normal development due to their continuous deposition onto layers of attached cells, hence, their significance in biofilm formation should be acknowledged. The flavonoid quercetin inhibited *L. monocytogenes* biofilm growth with an MBC of 0.8 mM, according to the findings. This dosage was six times lower than the concentration required to halt planktonic growth (4.9 mM), implying that quercetin interferes with biofilm formation mechanisms other than cell division [49]. However, biofilm development was affected by increasing quercetin levels as concentrations of 0.2 and 0.4 mM resulted in significant reductions (*p* < 0.05) in viable surface-associated cells of 1.96 and 3.21 Log_10_ CFU/cm^2^, respectively [49].

However, when quercetin inhibited the biofilm, the inhibitory effect was greater on R surfaces than on HG surfaces at MIC of quercetin (Figure 3 and Figure 4). The use of plastic cutting boards for processing and cooking raw foods is very susceptible to cross-contamination [50] because *Salmonella* can adhere to the surfaces of plastic to form a biofilm [11]. Additionally, because plastic is a hydrophobic material, *Salmonella* bacteria are more likely to adhere on it than on glass and stainless steel surfaces, which are hydrophilic materials [51]. Therefore, it is essential to prevent contamination of plastic cutting boards used when processing or cooking foods, which cause Salmonellosis. The ability of quercetin to prevent the production of biofilms in *Staphylococcus epidermidis* was investigated by other authors. Biofilm development was reduced by quercetin in a concentration-dependent way. At concentrations of 250 μg/mL and 500 μg/mL, quercetin inhibited *S. epidermidis* biofilm formation by 90.5% and 95.3%, respectively [30]. It was found that there were 13–72, 8–80, and 10–61% reductions in biofilm formation of three Gram-negative food-borne bacteria, *Klebsiella pneumoniae*, *P. aeruginosa*, and *Yersinia enterocolitica*, respectively, at varied doses of 5–40 μg/mL [52]. On stainless steel, a substantial reduction of 1.48 Log_10_ CFU/cm^2^ of *Listeria monocytogenes* biofilm population was measured when quercetin was present at 0.2 mM compared to the control [49]. When quercetin levels were increased to 0.4 and 0.8 mM, there were no visible living cells adhering to the test surfaces. After 24 h of incubation, cell densities in control biofilms increased to 6.09 Log_10_ CFU/cm^2^. Increasing quercetin levels, on the other hand, affected biofilm formation, with doses of 0.2 and 0.4 mM resulting in 1.96 and 3.21 Log_10_ CFU/cm^2^ reductions in viable surface-associated cells, respectively [49].

The quercetin produced demonstrated the ability to inhibit biofilm formation and eradicate established biofilms involving the production of reactive oxygen species (ROS), indicative of membrane activity [29,53]. The inhibitory effect of quercetin on the *S.* Typhimurium biofilm on the HG surface was visually confirmed with FE-SEM, and the results are shown in Figure 5. In HG surface, the bacteria of the control groups were gathered together and biofilms were formed around them. However, as the concentration of quercetin increased, the bacteria existed independently and biofilm formation was no longer observed. It was reported that quercetin inhibited *S.* Typhimurium biofilms, which is important to this study [54]. Biofilms treated with 125 μg/mL quercetin developed thinner and looser, and were significantly easier to remove than untreated biofilms. The SEM examination revealed that cells treated with quercetin adhered to the coverslips less. Additionally, the treated group had fewer intercellular chemicals than the untreated group. No morphologic changes were observed in the presence of quercetin. The growth curves of *S. epidermidis* cells were also examined in the presence of quercetin (125 μg/mL), and no decrease in cell growth was observed [30]. Previously reported [30], the prevention of *S. epidermidis* biofilm formation by quercetin was attributed to antibiofilm action rather than antibacterial activity, based on cell growth and microscopic data. In this study, the inhibition of biofilm was much greater in 1/2 than 1/8 MIC in Figure 5.

Many genes are important in *S.* Typhimurium physiological characteristics, biofilm development, QS, and pathogenicity. To assess the potency of quercetin, we examined the expression profiles of genes involved in QS, stress response, and virulence in *S*. Typhimurium (*rpoS*, *avrA*, *hilA*, and *luxS*). Pathogenecity, QS, virulence factors, and biofilm-forming processes are all linked. Prevention or inhibition of QS production is an emerging method for inhibiting biofilm development, reducing pathogenic infections, and ensuring food safety. When ROS accumulates inside the cell, it results in oxidative stress [15]. Oxidative stress plays an important role in biofilm formation by improving adaptability to microbial populations and protection for survival [31]. ROS are important signaling molecules not only in human cells, but also in microorganisms. ROS can act as both intracellular and extracellular stimulants to maintain a healthy redox cycle and to promote microbial attachment, consequently leading to the development of biofilms [31]. A disturbance in the redox cycle can lead to an accumulation. Quercetin is an antioxidant that disrupts biofilm formation by releasing ROS inside cells and damaging membrane integrity of bacterial cell [29]. The *rpoS* regulates the stationary-phase expression of a group of genes related to resistance to various environmental stresses [12]. These environmental stresses include low pH, starvation, temperature change, and oxidative stress [4]. In the present study, a significant reduction (*p* < 0.05) of *rpoS* gene expression was observed in the presence of quercetin at its 1/2 MIC. This result suggests that the oxidative stress of *Salmonella* was lowered by the free radical scavenging function of quercetin, and the expression of the *rpoS* gene was reduced. *Salmonella* pathogenicity islands (SPI–1) play an important role in the pathogenicity of *Salmonella* spp. as several virulence factors that induce epithelial cell invasion and macrophage dearth are gathered [55]. SPI-1 virulence genes include *hilA*, *hilC*, *hilD*, *sopB*, *sopD*, *sopE2*, *sipA*, *sipC*, *avrA*, and *sptP* [56]. One of them, *hilA*, encodes a transcriptional regulator and plays a key role in *S*. Typhimurium invasion [57]. *AvrA*, another SPI–1 virulence gene, plays an inhibitory role in inflammation, allowing the pathogen to survive well in the host [58]. In the present study, a significant reduction (*p* < 0.05) of *hilA* and *avrA* genes expression was observed in the presence of quercetin at its 1/2 MIC. Another study reported [48] that the gene expressions of *rpoS*, *luxS*, *hilA*, and *avrA* were significantly (*p* < 0.05) decreased at 1/2 MIC of quercetin against *S*. Typhimurium. *Pseudomonas aeruginosa* biofilm formation and virulence factor secretion were both inhibited by quercetin, and 16 μg/mL quercetin significantly reduced the expression of *lasI*, *lasR*, *rhlI*, and *rhlR* [33]. Given the importance of QS in controlling biofilm formation and virulence factor production, we hypothesized that quercetin-mediated reduction of biofilm formation and virulence factor production occur via effects on QS. Furthermore, QS is a suitable target for biofilm infection management approaches. The QS is a cell density-dependent mechanism that allows multicellular organisms to make collective decisions and synchronize with the rest of the population [59,60]. Signaling autoinducer-2 (AI–2), a putative quorum–sensing signal, is vital in biofilm development in many bacterial species [61]. Quercetin prevents biofilm formation in *S. epidermidis* by lowering EPS synthesis and modifying the composition of EPS [30]. *LuxS* synthesizes the AI–2 molecules and causes quorum-sensing. In this experiment, gene expression of *luxS* decreased significantly at the different levels of sub-MIC of quercetin (*p* < 0.05). All the downregulating genes of these experiments showed their reduced expression levels when tested with different sub–inhibitory concentrations of quercetin.

## 5. Conclusions

In the food industry, preventing biofilm formation is crucial for maintaining a high degree of food safety. In conclusion, we showed initial evidence that quercetin has an antibacterial effect on *S.* Typhimurium biofilms on food–contact surfaces. Quercetin can suppress the expression of QS, virulence, and stress response genes, in addition to preventing bacterial biofilm formation. This conclusion is especially significant, given the critical gap in food safety hazards observed in the majority of food additives investigated to date, such as quercetin, which was proven efficient against bacterial biofilm formation despite any noticeable detrimental effects on bacterial cells. Quercetin is a dietary component obtained from plants that is both cheap and effective. Given the problems that biofilm causes in the health and industrial sectors, creating effective control measures and using the right techniques to evaluate their effectiveness is crucial in the fight against biofilms. According to the findings of this study, quercetin can operate as a biofilm inhibitor, reducing *S.* Typhimurium biofilm development on food–contact surfaces in processing plants and the food industry.

## Figures and Tables

**Figure 1 foods-11-00977-f001:**
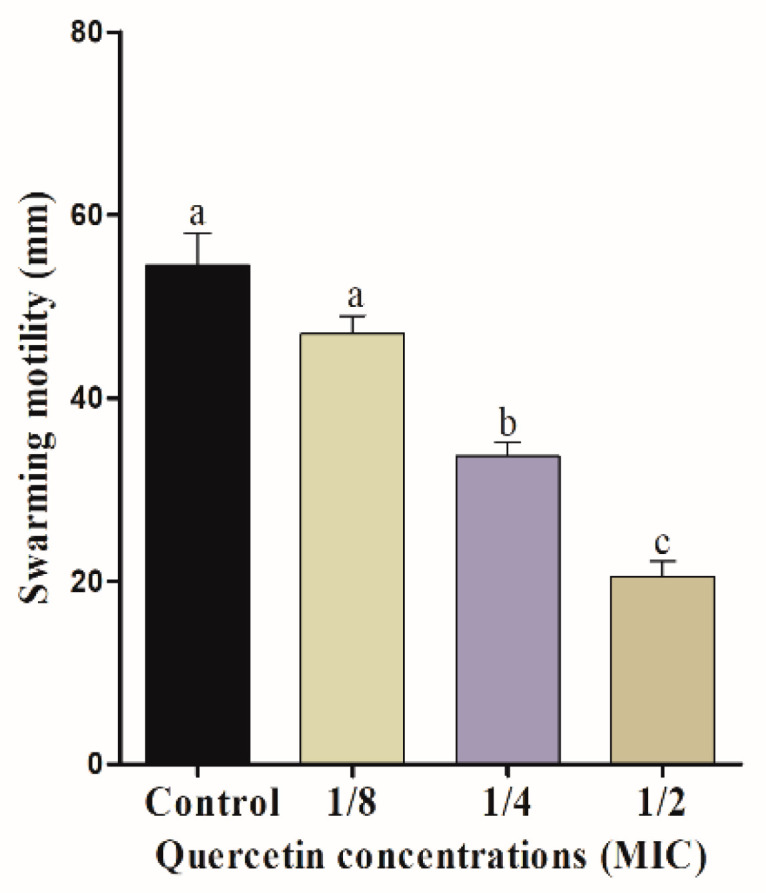
Swimming motility assay for *Salmonella* Typhimurium with various amount of quercetin (μg/mL). Data are expressed as mean ± SEM of three independent replicates. ^a–c^ Values with different letters are significantly different by Duncan’s multiple-range test (*p* < 0.05).

**Figure 2 foods-11-00977-f002:**
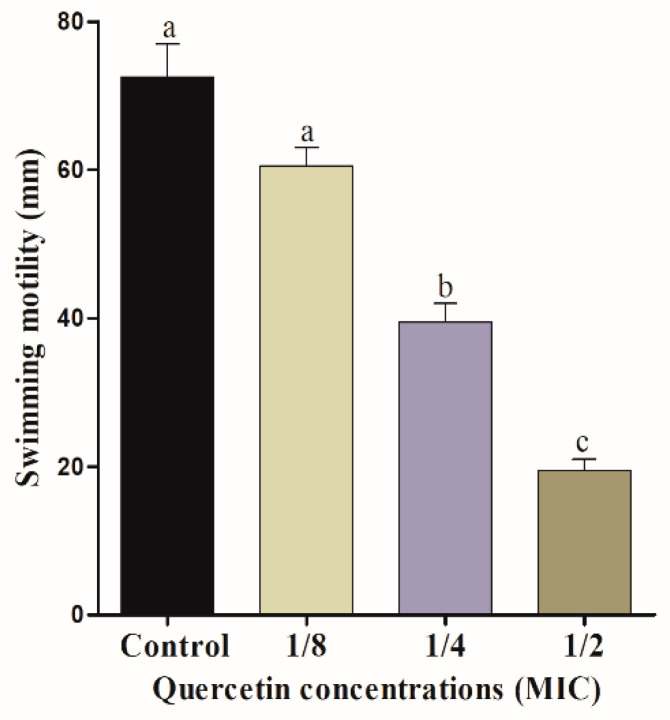
Swarming motility assay for *Salmonella* Typhimurium with various amount of quercetin (μg/mL). Data are expressed as mean ± SEM of three independent replicates. ^a–c^ Values with different letters are significantly different by Duncan’s multiple-range test (*p* < 0.05).

**Figure 3 foods-11-00977-f003:**
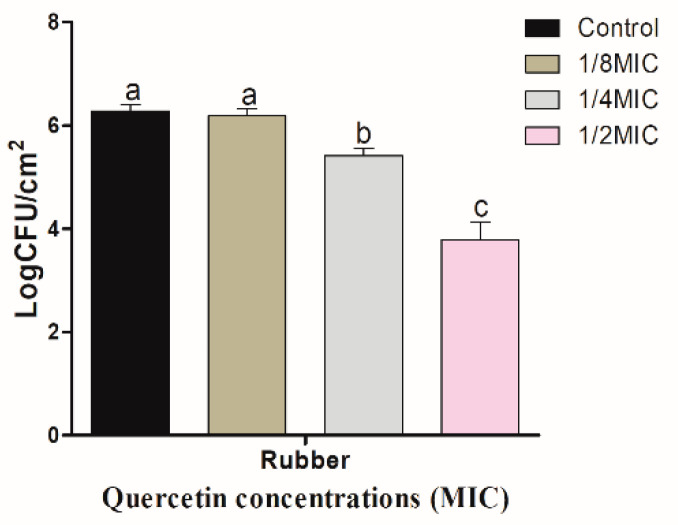
Inhibition of *Salmonella* Typhimurium biofilm formation (24 h) on rubber surfaces by various concentrations of quercetin (1/8, 1/4, and 1/2 MIC). Data are expressed as mean ± SEM of three independent replicates. ^a–c^ Values with different letters are significantly different by Duncan’s multiple-range test (*p* < 0.05).

**Figure 4 foods-11-00977-f004:**
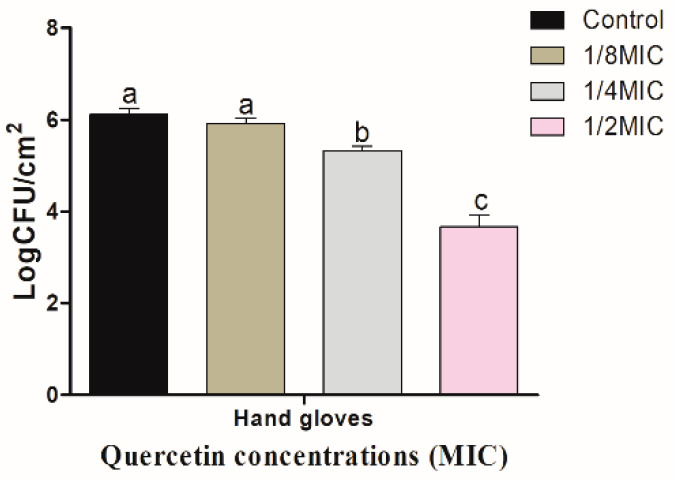
Inhibition of *Salmonella* Typhimurium biofilm formation (24 h) on hand gloves surfaces by various concentrations of quercetin (μg/mL). Data are expressed as mean ± SEM of three independent replicates. ^a–c^ Values with different letters are significantly different by Duncan’s multiple-range test (*p* < 0.05).

**Figure 5 foods-11-00977-f005:**
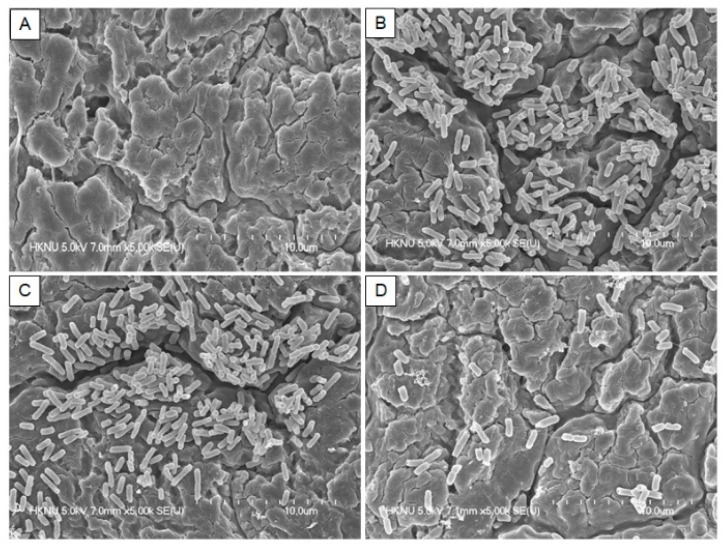
Representative scanning electron micrographs of *Salmonella* Typhimurium biofilms formation in the presence of various amounts of quercetin on the hand gloves surfaces. (**A**) Blank; (**B**) Control (0% quercetin); (**C**) 1/8 MIC; (**D**) 1/2 MIC.

**Figure 6 foods-11-00977-f006:**
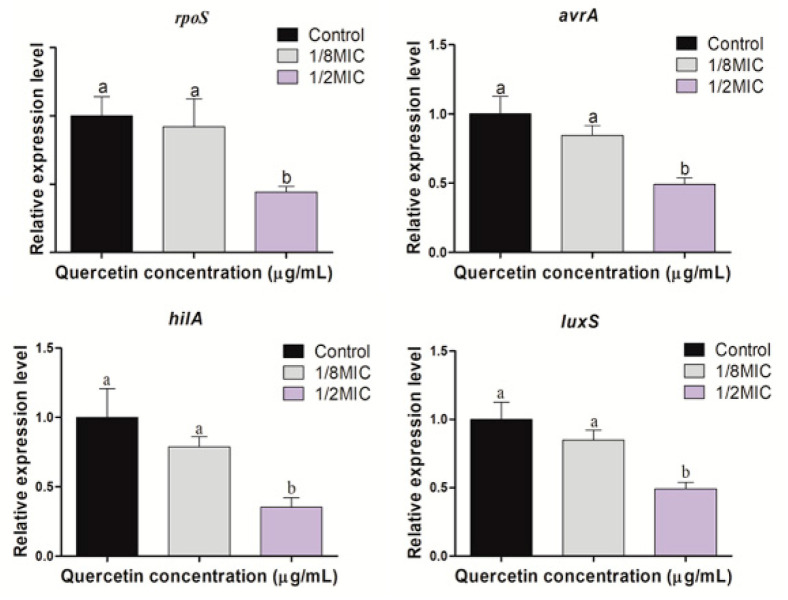
Relative expression levels of *rpoS*, *avrA*, *hilA*, and *luxS* genes in *Salmonella* Typhimurium suspension, supplemented with various amounts of quercetin. ^a,b^ Different superscript letters indicate significant differences (*p* < 0.05) with three independent replicates.

**Table 1 foods-11-00977-t001:** List of primers used in this study that are related to virulence, stress response, and quorum sensing genes. F and R stand for forward and reverse primers, respectively.

Target Primers	Sequence (5′-3′)	Product Size (bp)
16S rRNA	F: CAGAAGAAGCACCGGCTAAC R: GACTCAAGCCTGCCAGTTTC	167
*rpoS*	F: GAATCTGACGAACACGCTCA R: CCACGCAAGATGACGATATG	171
*avrA*	F: GAGCTGCTTTGGTCCTCAAC R: AATGGAAGGCGTTGAATCTG	173
*hilA*	F: ATTAAGGCGACAGAGCTGGA R: GCAGAAATGGGCGAAAGTAA	134
*LuxS*	F: CGGGTTGCAAAAACGATGA R: GTTGAGGTGGTCGCGCATA	150

## Data Availability

Data is contained within the article.

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
