# Peer review of "Impact of Quercetin against *Salmonella* Typhimurium Biofilm Formation on Food–Contact Surfaces and Molecular Mechanism Pattern"

_foods, 2022, doi:10.3390/foods11070977_

Round 1
Reviewer 1 Report
Impact of quercetin against Salmonella Typhimurium biofilm formation on food–contact surfaces and molecular mechanism pattern
In this study, authors used quercetin to inactivate the biofilm of Salmonella Typhimurium on food contact surface. The results are impressed. However, concentrations of stable and measurable RONS, such as nitrate, nitrite, hydrogen peroxide, and ozone were not provided. Authors also did not provide statistical significance for the results. Some description of device design was also missing.
- Title: Salmonella Typhimurium Typhimurium should not be italic.
- Line 86 and 89: bacterial name should be italic.
- Line 110: what was the materials of hand glove? Nitrile? Latex? Power or non-power?
- Line 122: what was the wavelength of absorbance?
- Line 143: was 24-h incubation enough to form biofilm?
- Line 167-171: The bacterial cells used for RNA extraction were more likely to be planktonic cells according to this description. Was it suitable to present the situation of bacterial cells in biofilm?
- Line 183: was 50°C annealing time 20 s?
- Line 194: Typhimurium, S was not italic. There are several non-italic bacterial names. Recommend authors checking throughout the manuscript.
- Line 226: cm2, no superscript
- Line 251: what was the gene expression when 1/4 MIC used?
- Gene expression section: Suggest authors describing more about the gene expression and biofilm inactivation.
- Fig 5. How were the SEM results on rubber? Were they similar to hand glove?
- Discussion: although quercetin is a natural extract, what is its toxicological status? Is quercetin on GRAS list?
- Line 273: since swarming is the key factor for biofilm formation, suggest authors discussing more the indication of swarming results.
- Line 276: “The greater the inhibitory impact of quercetin, the higher the concentration.” Would it be better “The higher the concentration, the greater the inhibitory impact of quercetin.”
- Line 280-286: this paragraph mentioned two concentrations, 0.2 mM and 0.8 mM for the study of monocytogenes. Besides, the description of this paragraph confuse me.
- Line 294: “are more susceptible to adhesion” do authors mean more likely to adhere?
- Line 297: non-italic bacterial name. Recommend authors to check throughout this manuscript.
- Line 304: Log10 CFU/cm2 : 10 should be subscript. Recommend authors to check throughout this manuscript.
- Line 334: Usually, ROS possesses a negative effect to bacterial growth. However, authors describe ROS is able to stimulate the biofilm formation. Could authors provide more evidence or references for ROS inhibiting the growth of planktonic cells but stimulating formation of biofilm?
Reviewer 2 Report
Suggestions of minor corrections are stated in attached file.

Reviewer 3 Report
The proposal of the work presented is of great relevance and the manuscript is very well written. The results were presented in a simplified and easy-to-understand way for the reader. Below are a few considerations and observations:1. There is no need for native English language correction for the manuscript as the text is well written. However, in item 2.4. "Determination of Minimum Inhibitory Concentration (MIC)", there is limited use of words such as, "So" and "Then" in a single paragraph of only seven lines. This also appears in other paragraphs throughout the manuscript on material and methods. I recomend the use of the other gramatical uses.
2. The experiments performed were performed for the main question presented by the authors and I believe that no additional experiments were performed.
3. The quality of the figures is good, requiring no additional adjustments or edits.
